# Evolutionary trade-offs in dormancy phenology

**Théo Constant[1], F Stephen Dobson[1,2], Caroline Habold[1]\*[†], Sylvain Giroud[3,4]\*[†]**

[1]UMR 7178, Centre National de la Recherche Scientifique, Institut Pluridisciplinaire Hubert CURIEN, Université de Strasbourg, Strasbourg, France; [2]Department of Biological Sciences, Auburn University, Auburn, United States; [3]Department of Interdisciplinary Life Sciences, Research Institute of Wildlife Ecology, University of Veterinary Medicine Vienna, Vienna, Austria; [4]Energetic Lab, Department of Biology, Northern Michigan University, Marquette, United States

**\*For correspondence:**
caroline.habold@iphc.cnrs.fr (CH);
sgiroud@nmu.edu (SG)

[†]These authors contributed equally to this work

**Competing interest:** The authors declare that no competing interests exist.

**Abstract** Seasonal animal dormancy is widely interpreted as a physiological response for surviving energetic challenges during the harshest times of the year (the physiological constraint hypothesis). However, there are other mutually non-exclusive hypotheses to explain the timing of animal dormancy, that is, entry into and emergence from hibernation (i.e. dormancy phenology). Survival advantages of dormancy that have been proposed are reduced risks of predation and competition (the 'life-history' hypothesis), but comparative tests across animal species are few. Using the phylogenetic comparative method applied to more than 20 hibernating mammalian species, we found support for both hypotheses as explanations for the phenology of dormancy. In accordance with the life-history hypotheses, sex differences in hibernation emergence and immergence were favored by the sex difference in reproductive effort. In addition, physiological constraint may influence the trade-off between survival and reproduction such that low temperatures and precipitation, as well as smaller body mass, influence sex differences in phenology. We also compiled initial evidence that ectotherm dormancy may be (1) less temperature dependent than previously thought and (2) associated with trade-offs consistent with the life-history hypothesis. Thus, dormancy during non-life-threatening periods that are unfavorable for reproduction may be more widespread than previously thought.

## eLife assessment

This **valuable** and ambitious review examines seasonal dormancy in various species, including hibernating mammals (excluding bats and bears) and ectotherms. It provides a **solid** test of hypotheses on dormancy timing, considering energetic constraints and life history as alternative drivers. The review will be of interest to evolutionary biologists.

## Introduction

A large number of species across the tree of life enter prolonged dormancy each year (*Wilsterman et al., 2021*). From a physiological point of view, dormancy occurs under a combination of high energy reserves and a significant reduction in energy demand, thus allowing prolonged inactivity for several months to several years (*Hoehler and Jørgensen, 2013*; *Staples, 2016*). From an evolutionary perspective, dormancy improves survival during stressful environmental periods, until subsequent conditions that favor reproduction (*Watts and Tenhumberg, 2021*). The evolutionary tactic of dormancy has been studied independently among different major phylogenetic groups (*Wilsterman*

*et al., 2021*), and these separate considerations have limited the development of a global evolutionary framework to explain dormancy.

The dormancy of plants, micro-organisms, and some invertebrates has been extensively studied from an evolutionary point of view. In these species and in addition to energetic benefits, dormancy occurs in a large number of situations that reduce competition and predation (*Danks, 1992*; *Satterthwaite, 2010*; *Blath and Tóbiás, 2020*). The evolution of dormancy is explained as based, for example, on 'evolutionarily stable strategies' (*Hairston, and Munns,, 1984*; *Kortessis and Chesson, 2019*), 'life-history theory' (*Ji, 2011*; *Watts and Tenhumberg, 2021*), or as a 'bet-hedging strategy' (*Hopper, 1999*; *Joschinski and Bonte, 2021*).

In most animals, however, the topic of prolonged dormancy has primarily focused on ecophysiology rather than evolutionary biology. Dormancy duration increases with physiological constraints in the environment (e.g. energy shortage, thermic, or water constraints), demonstrating its adaptive role in response to short growing season (*Pianka, 1970*; *Turbill and Prior, 2016*; *Wilsterman et al., 2021*). However, some animals immerge into dormancy while environmental conditions would allow (from a physiological point of view) a continuation of activity, suggesting other survival benefits than coping with a short growing season (*Jameson and Allison, 1976*; *Wiklund et al., 1996*; *Humphries et al., 2002*). A reduction in the risk of predation or competition during animal dormancy has been suggested based on increased survival during hibernation, compared to the active season (*Turbill et al., 2011*; *Ruf, 2012*; *Constant et al., 2020*), in particular from studies of the hibernating edible dormouse (*Glis glis*) (*Bieber and Ruf, 2009*; *Bieber et al., 2014*; *Hoelzl et al., 2015*; *Ruf and Bieber, 2023*). To date, however, the generality of an influence of these factors on the evolution of prolonged dormancy lacks attention. This raises the question of whether the timing of animal dormancy, that is dormancy phenology, might be exclusively explained by physiological constraint or whether other ecological factors may be involved.

Among dormant animals, a general distinction is made between heterothermic endotherms (some mammals and birds) that are able to actively influence their metabolic rate via oxidative metabolism and ectotherms (invertebrates, fish, amphibians, and reptiles) whose metabolic rates are more subject to microclimatic fluctuations (*Staples, 2016*). In both cases of heterothermy, although dormancy drastically reduces energy expenditure, some energy is nonetheless lost in the absence of an external source. As a consequence, if dormancy phenology is explained solely by physiological issues (the physiological constraint hypothesis), selection should favor remaining active until a positive energy balance (in endotherms) or thermal window favorable for activity (in ectotherms; see *Gunderson and Leal, 2016*) is no longer possible (*Figure 1a*). If, however, there are other benefits to dormancy such as improved survival due to a reduction of predation risk, these survival benefits may produce a trade-off between being active and investing in reproduction *versus* being dormant for a time to increase survival (the life-history hypothesis). Nevertheless, these two hypotheses are not mutually exclusive, since harsh climatic conditions and low food availability may reduce the benefits of being active for reproductive success and thus influence this trade-off. Within species, this trade-off may be reflected by sex differences in the phenology of dormancy, since males and females exhibit differences in reproductive timing and investment (*Emlen and Oring, 1977*).

In a recent study, predation avoidance and sexual selection received support for explaining intraspecific variation in hibernation phenology in the northern Idaho ground squirrel (*Urocitellus brunneus*, *Allison et al., 2023*) Males often emerged from dormancy and arrived at mating sites some days or weeks before females (termed 'protandry'), and mating occurred shortly after female emergenced from dormancy. Sexual selection may favor a life history in which relatively early-emerging males benefit from greater reproductive success (the 'mating opportunity hypothesis,' *Morbey and Ydenberg, 2001*). Males that are physiologically prepared to mate (*Breedveld and Fitze, 2016*) and have established intrasexual dominance or territories (*Manno and Stephen Dobson, 2008*; *Hibbitts et al., 2012*) prior to mating are likely to have greater reproductive success (*Michener, 1983*; *Michener, 1984*). Thus, greater protandry is assumed to have evolved with intraspecific competition and longer periods of mating preparation. For females, emergence phenology may promote breeding and/or care of offspring during the most favorable annual period (e.g. a match of the peak in lactational energy demand and maximum food availability, *Figure 1*) or beginning early to afford long active seasons for offspring while not compromising the survival of parents. Although males are active above ground before females, the latter sex may not emerge until later to limit mortality risks (see the

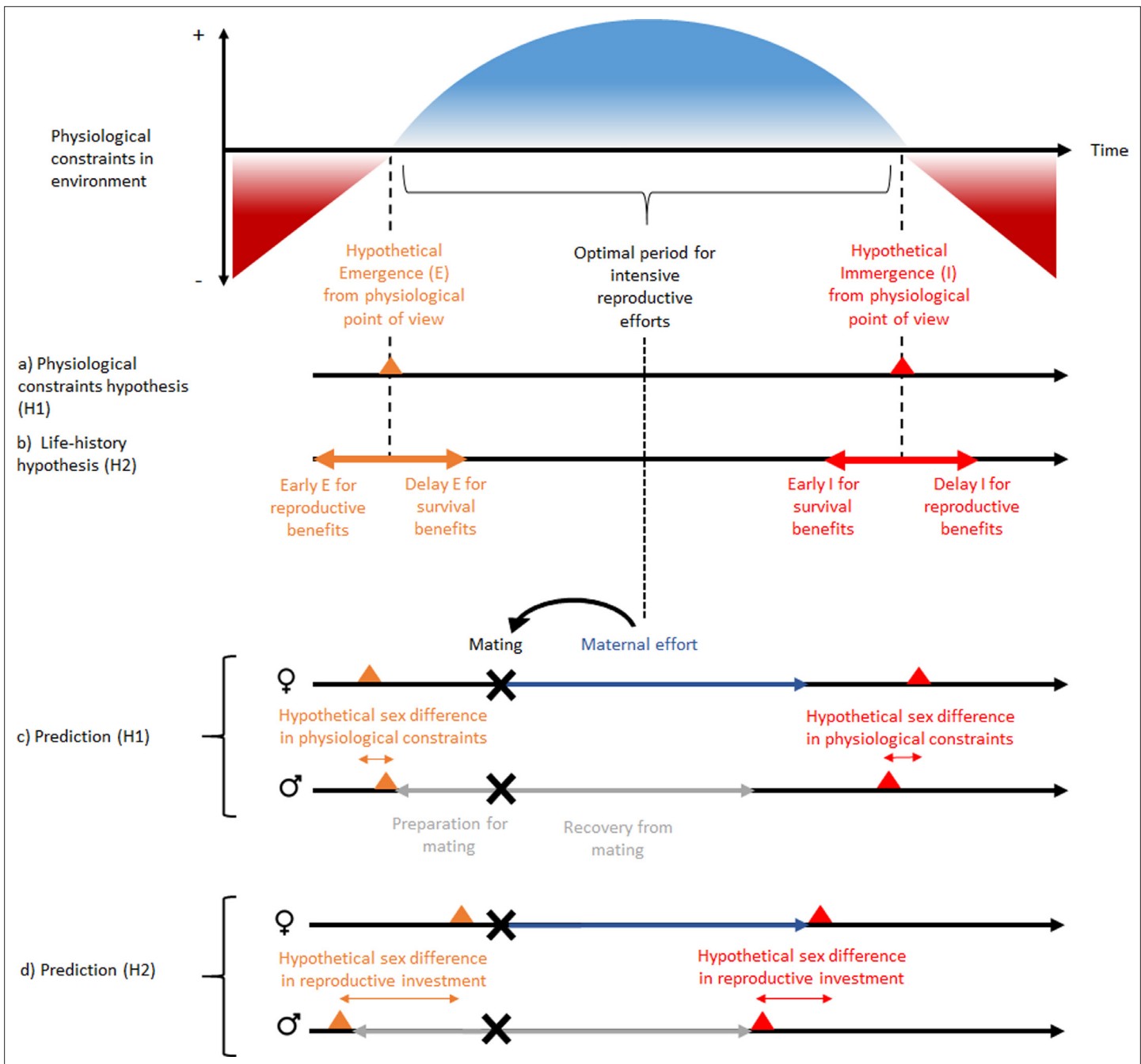

**Figure 1.** Schematic representation of 'the physiological constraint hypothesis' and 'the lifehistory hypothesis' and their predictions. Physiological constraints in the environment refers to variation over time of the energetic balance (mainly for heterothermic endotherms) and the thermal window favorable to activity (mainly for ectotherms). Hypothesis H1 assumes that dormancy phenology occurs at the time of transition between favorable and unfavorable energetic or thermal conditions or *vice versa*. It predicts that the sex difference in dormancy phenology is explained by sex differences in physiological constraints, and that reproductive investment should be independent of this sex difference in phenology. In contrast, hypothesis H2 predicts that a phenology that would occur before or after the energetic transition may be associated with benefits to survival or reproduction. It predicts that the sex difference in dormancy phenology is associated with a sex difference in reproductive investment. This pattern is expected for species without paternal effort. But the concept can be applied to other types of mating strategies. The hibernation phenology presented for prediction (**H1**) are those expected for hibernating mammals. Note that the magnitude and order of sex differences in phenology is not an expected general trend, because it is assumed to vary between species according to beneficial envi600ronmental conditions for energy demand (prediction H1) and reproductive investment (prediction H2). Nevertheless, the sex difference is assumed to be smaller with the H1 prediction because there is less sex difference in energy demand than in reproductive investment. Black, gray, and dark blue horizontal arrows represent respectively time over the year, reproductive investment in males and reproductive investment in females. Black, gray, and dark blue horizontal arrows represent respectively time over the year,

*Figure 1 continued on next page*

*Figure 1 continued*

reproductive investment in males and reproductive investment in females. Red and orange triangles represent respectively immergence and emergence timing and black cross represent mating timing.

'waiting cost hypothesis,' *Morbey and Ydenberg, 2001*). Later in the year, both sexes are expected to enter dormancy when they are no longer investing in or recovering from reproduction, and after acquiring adequate energy stores for overwintering.

In the present study, we investigated the 'physiological constraints' and 'life-history' hypotheses, two non-mutually exclusive evolutionary explanations for the phenology of dormancy, especially in regard to sex differences within species. To examine these hypotheses, we used two complementary approaches: (i) a set of phylogenetic comparative analysis, and (ii) a comparison between endotherm and ectotherm dormancy via the conducted analysis (from endotherms, including mainly holarctic rodents) and the existing literature (from ectotherms).

First, using phylogenetic comparative analysis, we compared traits associated with climate, reproduction, body mass, and hibernation to explain sex differences in phenology in more than 20 hibernating mammals (*Figure 2*). To begin with, we predicted from the life history hypothesis that (1) males of species with high energetic cost of reproduction would exhibit greater protandry due to longer mating preparation. We tested whether body mass lost during mating was associated with greater protandry. For immergence, males were expected to be constrained by the long-term negative effects of reproductive stress (*Millesi et al., 1998*), whereas female immergence was constrained by maternal effort (*Levesque et al., 2013*). Thus, we expected that (2) the species with higher sex differences in energetic cost of reproduction would have a greater sex difference in the timing of immergence, with the sex that invested more energy or time in reproduction also being the one that immerged last. If sex differences in hibernation phenology were solely explained by the physiological constraint hypothesis, then we would predict that species with high sexual dimophism (proxy of sex difference in energetic demand) in body mass would exhibit greater sex differences in hibernation phenology. On the contrary, if physiological constraints influenced the trade-off between survival and reproduction, then we would predict that species living in short growing seasons (low temperature and precipitation)

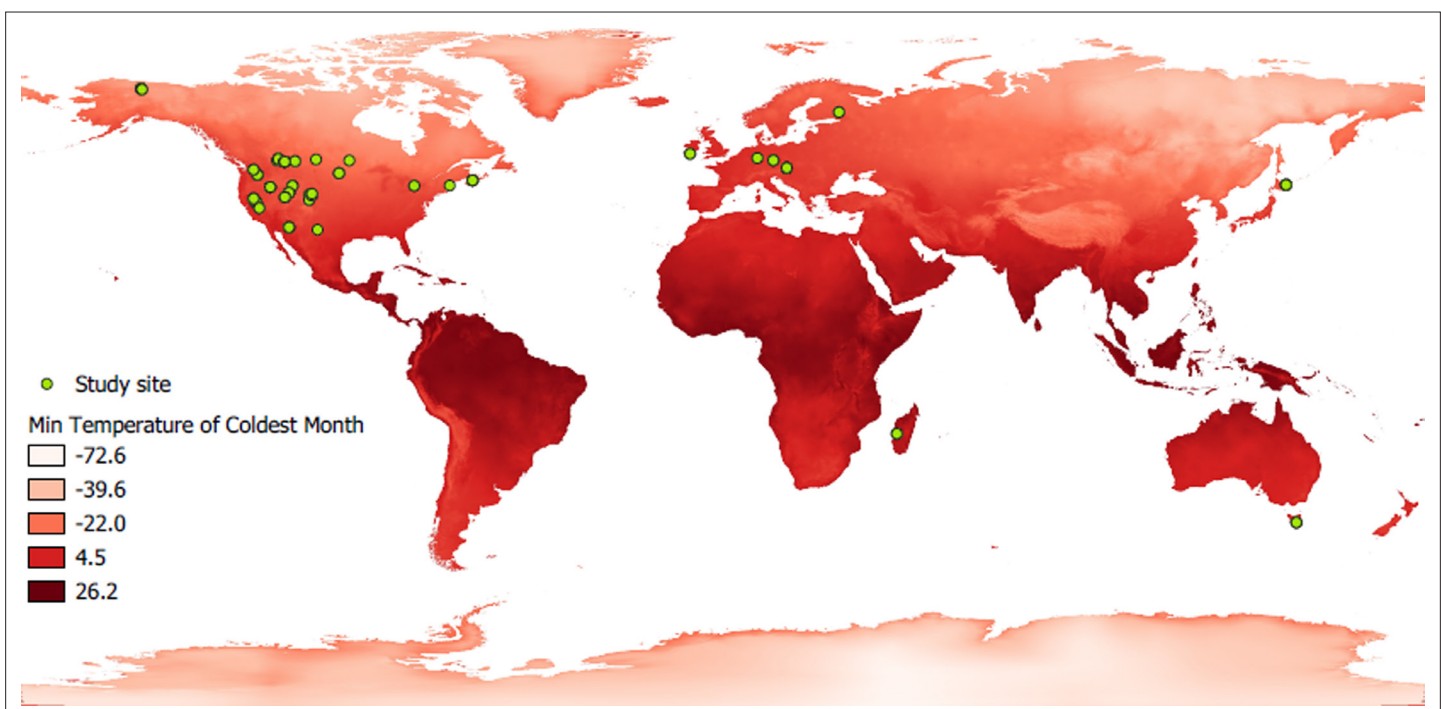

**Figure 2.** World map showing the study sites of the hibernating species studied. Green dots cindicate the location of study sites. A red gradient represents the variation in minimum ambient temperature, a parameter used in this study.

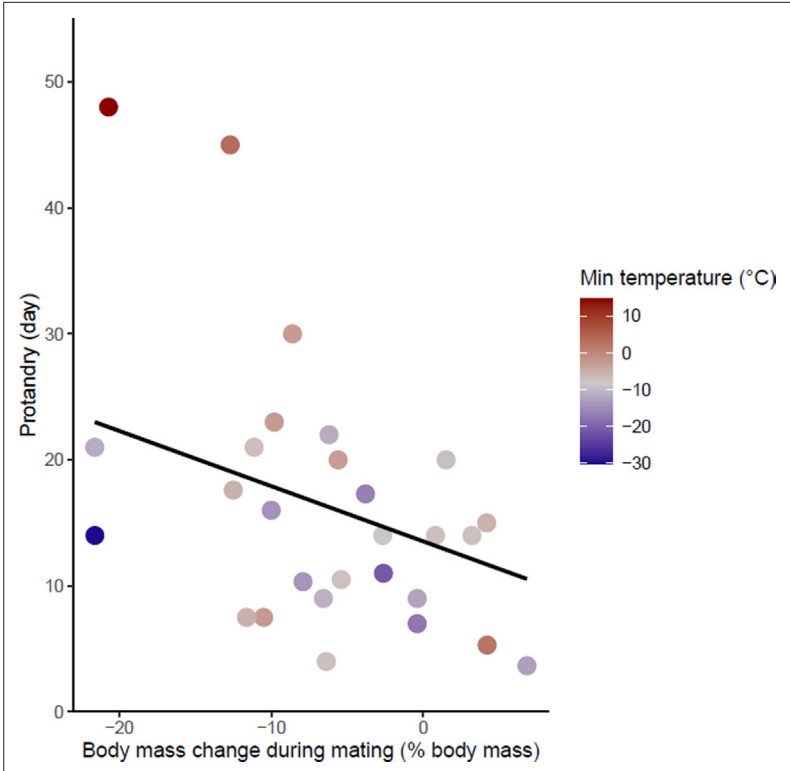

**Figure 3.** Effects of body mass variation during mating on protandry. The minimum temperatures of the study sites are indicated by a color gradient with the warmest temperatures in red. The regression line in black indicates the effect of body mass variation during mating on protandry. Body mass variation during mating was represented as a percentage of body mass before mating.

The online version of this article includes the following source data for figure 3:

**Source data 1.** Hibernation phenology, reproduction, body mass and climate data used in Model 1.

and smaller species would exhibit lower sex differences in hibernation phenology due to higher cost of thermoregulation during activity (*Figure 3*, *Figure 4*, *Figure 5*, *Figure 6*).

Second, there is a need to unify the study of dormancy across ectotherms and endotherms for answering fundamental evolutionary questions (*Wilsterman et al., 2021*). While there are insufficient data on dormancy phenology and reproductive investment in ectotherms to allow a statistical comparative analysis, we examined, as a second step, the relationships between reproductive investment, energy balance, thermal constraints, and dormancy phenology of ectotherms that are already available in the literature. We compared these studies for ectotherms to our results for endotherms. Finally, we highlighted evidence from the literature for the independence of dormancy phenology from energy balance.

## Results

### Emergence

Protandry increased significantly with male body mass loss during mating and increased ambient temperature (model 1 in *Table 1* and *Figure 3—source data 1*). At the same time, protandry decreased in species for which males are smaller. The effect of ambient temperature on protandry is almost two times greater than male body mass and body mass loss during mating. There is also a trend towards lower protandry with increasing time between female emergence and the start of mating but this relationship is not significant ('late mating' in model 1, *Table 1*). This model showed no influence of phylogeny.

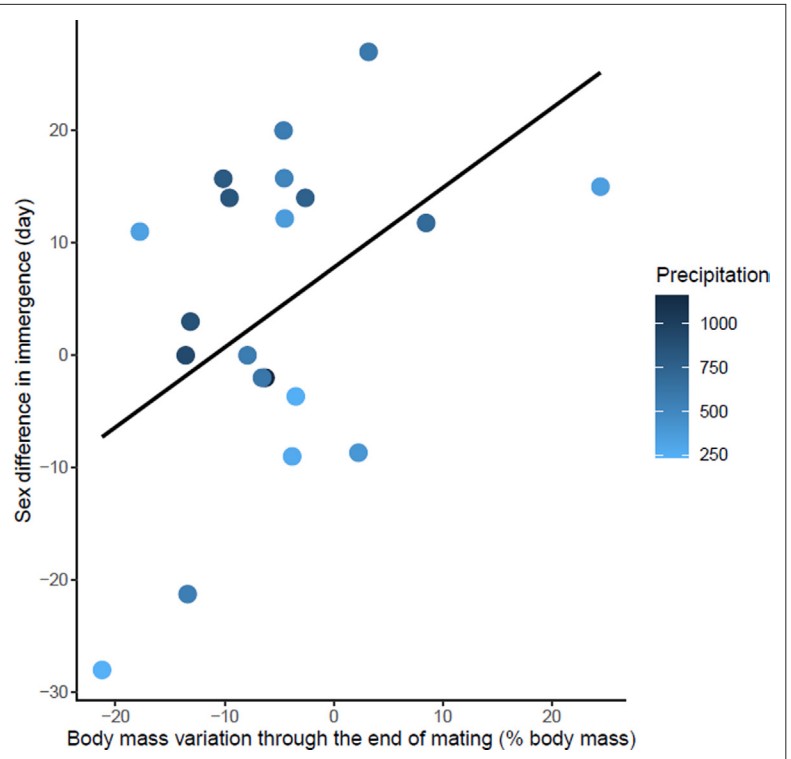

**Figure 4.** Effects of body mass variation through the end of mating on sex differences in immergence. Annual precipitation at the study sites are indicated by a color gradient with the highest precipitation in dark blue. The regression line in black indicates the effect of body mass variation through the end of mating on sex difference in immergence. Body mass variation through the end of mating was represented as a percentage of body mass at emergence. Males immerged earlier in species living in an environment with high precipitation (dark blue dots).

The online version of this article includes the following source data for figure 4:

**Source data 1.** Hibernation phenology, reproduction, body mass and climate data used in Model 2.

## Immergence

The sex difference in immergence date was associated with maternal effort duration, specific reproductive effort of female, Δ body mass through the end of mating, and precipitation (model 2, *Table 1*). High reproductive effort for both males and females delayed their immergence (*Figure 4—source data 1* and *Figure 5—source data 1*). At the same time, males immerged before females for species living in an environment with high annual precipitation. The effect of maternal effort duration was influenced by an outlier, *Tachyglossus aculeatus*, a species for which gestation and lactation for females was an extremely long period (i.e. 167 days). Maternal effort duration was not conserved in the best model when the outlier was removed (model 3, *Table 1*). There is also a trend towards early male immergence in species with male biased body mass dimorphism at immergence but this relationship is not significant (model 3, *Table 1*). Finally, phylogeny had no influence on this model.

## Discussion

### Sex difference in dormancy phenology

In this study, we investigated two mutually non-exclusive hypotheses about dormancy phenology: the physiological constraints and life history hypotheses. The sex difference in hibernation phenology is a good opportunity to confront these hypotheses, because the sexes are faced with somewhat different life-history challenges. The life-history hypothesis predicts a trade-off between the survival benefit of hibernation and the reproductive benefit of activity, such that the sex with lesser investment of time and energy in reproduction should spend more time in hibernation. If the physiological constraints hypothesis was the sole explanation for hibernation phenology, then species with

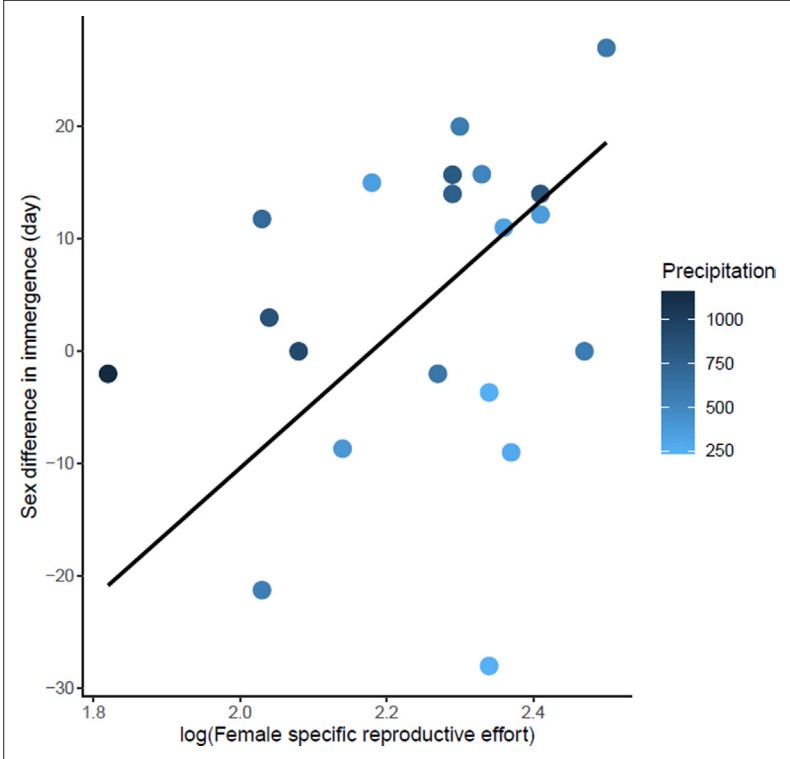

**Figure 5.** Effects of female specific reproductive effort on sex difference in immergence. The annual precipitation of the study sites are indicated by a color gradient with the highest precipitation in dark blue. The regression line in black indicates the effect of female specific reproductive effort on sex difference in immergence. Males immerged earlier in species living in an environment with high precipitation (dark blue dots).

The online version of this article includes the following source data for figure 5:

**Source data 1.** Hibernation phenology, reproduction, body mass and climate data used in Model 3.

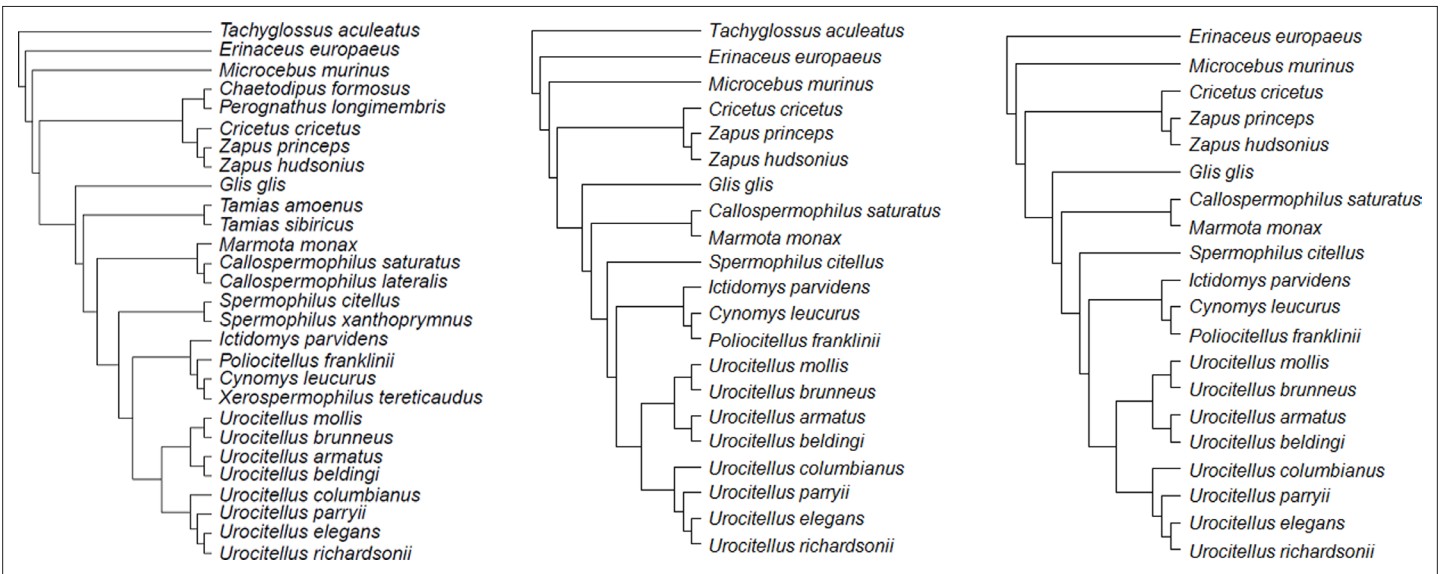

**Figure 6.** Consensus phylogenetic trees for the species under study: (**a**) model 1 (**b**) model 2, and (**c**) model 3. Each consensus tree was built from 100 trees obtained from http://vertlife.org/phylosubsets/. Branch lengths were calculated using Grafen's computations with the 'ape' package in R (see Materials and methods).

**Table 1.** Regression results for the best models explaining variation in protandry and sex difference in immergence. The Z standardized model estimates and the phylogenetic effect are reciprocally estimated by β and $\gamma_M$. The abbreviations 'diff' and 'rel' 'Min' stand respectively for 'difference,' 'relative,' and 'Minimum.' A negative value for the sex difference in immergence indicates that males immerge before females and a positive value indicates that females immerge before males. Relative testes mass, Δ body mass before mating, Δ body mass during mating, Δ body mass through the end of mating was represented respectively as a percentage of body mass, body mass at emergence, body mass before mating, and body mass at emergence.

| | R² | $\gamma_{ML}$ | Dependent variable | Independent variable | β±SE | t-value | p-value |
|---|---|---|---|---|---|---|---|
| Model 1 (28 species) | 0.59 | 0.00 (NA, 0.69) | Protandry | Intercept | 16.31±1.27 | 12.76 | <0.001*** |
| | | | | z- Male body mass at emergence | 3.25±1.37 | 2.37 | 0.02* |
| | | | | z-D body mass during mating | −3.32±1.39 | −2.38 | 0.02* |
| | | | | z-Min temperature | 5.94±1.36 | 4.34 | <0.001*** |
| | | | | z- late mating | −2.58±1.41 | −1.83 | 0.08. |
| Model 2 (21 species) | 0.50 | 0.00 (NA, 0.38) | Sex diff in immergence | Intercept | 5.18±2.18 | 2.36 | 0.03* |
| | | | | z- log specific reproductive effort | 13.77±3.88 | 3.54 | 0.002** |
| | | | | z-D body mass through the end of mating | 9.07±2.37 | 3.81 | 0.001** |
| | | | | z- precipitation | 7.67±2.59 | 2.95 | 0.009** |
| | | | | z- Maternal effort duration | 12.66±3.58 | 3.52 | 0.002** |
| Model 3 (20 species) | 0.63 | 0.00 (NA, 0.73) | Sex diff in immergence | Intercept | 4.24±1.89 | 2.23 | 0.04* |
| | | | | z- log specific reproductive effort | 10.20±2.34 | 4.35 | <0.001*** |
| | | | | z-D body mass through the end of mating | 7.05±2.05 | 3.42 | 0.003** |
| | | | | z- precipitation | 12.48±2.70 | 4.61 | <0.001*** |
| | | | | z-dimorphism at immergence | 4.75±2.42 | 1.96 | 0.06. |

high sexual dimorphism (proxy for energy demand) should show a large sex difference in hibernation phenology. As well, if physiological constraints influence the trade-off between survival and reproduction, then small species and species living in short growing seasons should show lower sex differences due to the higher costs of being active. Based on phylogenetic comparative analysis, we found support for both hypotheses. In accordance with the life history hypothesis, sex difference in emergence and immergence seem to reflect sex differences in reproductive effort. The comparative method, however, did not allow the assignment of causation of one variable on the other; that is, of the causal direction of selection pressures between reproductive investments and hibernation phenology. In addition, physiological constraint may influence the trade-off between survival and reproduction such that, low temperature and precipitation as well as smaller body mass influence sex differences in phenology.

Males with higher body mass loss during mating showed higher protandry. The males of some species increase locomotion and are subject to strong competition during mating, which greatly reduces their foraging time (*Millesi et al., 1998*). Thus, an early emergence of males may have evolved in response to sexual selection to accumulate energy reserves in anticipation of reproductive efforts. Females, on the contrary, are not subject to intraspecific competition for reproduction and may have sufficient time before (generally one week after emergence) and during the breeding period to improve their body condition. Protandry may also allow the maturation of the reproductive system or monopolization of territories (*Barnes et al., 1988*; *Millesi et al., 2008*; *Thompson et al., 2023*). In addition, protandry decreases in species living in cold environments and in species with lower male body mass in accordance with the physiological constraints hypothesis. Despite the reproductive benefits of early emergence, the cost of activity may be too high for these species, probably due to excessive heat loss in a cold environment.

In some species, males may have adaptations that improve their physical condition before reproduction without early emergence in cold climates. For example, males of some fat-storing species (e.g. *Urocitellus Parryii)* hoard food in their burrows. This energetic supply would support a return to euthermia of up to a few weeks prior to behavioral emergence and allow for testes maturation and fat accumulation (*Michener, 1992*; *Williams et al., 2014*), while remaining sheltered in the burrow. Thus, males might gain energy benefits without paying the survival costs of above-ground activity (*Turbill et al., 2011*; *Constant et al., 2020*). Although food storage in the burrow may have evolved to overcome short growing seasons or predation, model selection did not retain the food-storing factor. Thus, the ability to accumulate food in the burrow was not by itself likely to keep males of some species from emerging earlier (e.g. *Cricetus cricetus*, protandry: 20 days, *Siutz et al., 2016*). Early emerging males may benefit from consuming higher quality food or in competition with other males (e.g. dominance assertion or territory establishment, *Manno and Stephen Dobson, 2008*; *Thompson et al., 2023*).

There is a trend for decreased protandry in species for which reproduction occurs several weeks after female emergence, as demonstrated in lizards and snakes (*Graves and Duvall, 1990*; *Olsson et al., 1999*). According to the life history hypothesis, the benefits for males to emerge before females decreases with the delay in the mating period (relative to female emergence), because they are less constrained by time for mating preparation. This phenomenon concerns only a few species in our comparative analysis, which may explain its non-significant effect.

In ectotherms, testes maturation has been proposed as a major influence on protandry. In *Zootoca vivipara* (a viviparous lizard), male lizards generally emerged earlier than females (*Breedveld and Fitze, 2016*). The sex difference did not seem to be explained by a difference in the maturation duration of the reproductive organs. In this species, females did not have developed follicles at emergence and ovulation occurred several weeks after mating (*Bauwens and Verheyen, 1985*). In addition, by experimentally manipulating the emergence from dormancy of males but not females, it was shown that the degree of protandry affected the order of sperm presence in males, but not the probability of copulation. Thus, protandry may have increased the chances of fertilizing eggs for males and decreased the probability of copulating with an infertile male for females (*Breedveld and Fitze, 2016*). In *Gonepteryx rhamni* (the common brimstone butterfly), males emerged from dormancy 3 weeks before females. Males were quickly ready to reproduce, but this delay would have allowed them to increase their amount of sperm before mating and thus reproduce more successfully (*Wiklund et al., 1996*).

Unlike emergence, the sex that immerges into hibernation first varies among species. For males, high body mass loss before and during mating was associated with a delay in male immergence (for the same date of female immergence). In *Spermophilus citellus* (the European ground squirrel), the most actively mating males delay the onset of post-mating accumulation of body mass and also delay hibernation, presumably due to the long-term negative effects of reproductive stress (*Millesi et al., 1998*). For females, the higher the specific reproductive effort, the later the females immerged for the same date of male immergence. As for males, high reproductive investment by females may induce stress that delays immergence. The duration of maternal effort may also delay entry into hibernation, but its influence seems to result from an extreme value (*Tachyglossus aculeatus,* maternal effort duration:167 days). Therefore, it was the sex difference in the time and energy invested in reproduction which was associated with the order of immergence. As for emergence, physiological constraints due to short growing seasons may influence immergence. High precipitation seem to favor early immerge of males compared to females. High precipitation may be associated with high water and food availability enable a fast recovery from reproductive effort. However, females are constrained by maternal effort duration (gestation and lactation) which may result in early immergence of males compared to females. On the contrary, low precipitation may constrain recovery, and thus delay male immergence to a greater extent than females in comparison with an environment with high precipitation (in both environments, females are constrained by maternal care). In ectotherms, very few data are available on sex differences in immergence date and therefore do not allow evaluation of this hypothesis.

Although the sex difference in dormancy phenology seems to be widely supported for a trade-off between survival and reproduction, evidence exists at other scales of life. In mammals, males and females that invest little or not at all in reproduction exhibit advances in energy reserve accumulation and earlier immergence for up to several weeks, while reproductive congeners continue an activity (*Neuhaus, 2000*; *Millesi and Divjak, 2008*). Another surprising example of this trade-off is *Glis glis* (the edible dormouse), for which emergence became earlier with age. The authors posited that, as younger individuals had a greater chance of reproducing in subsequent years, they delayed their emergence for survival benefits at the expense of their immediate reproductive success (*Bieber et al., 2018*). In ectotherms, several studies suggest that the benefits for reproduction (*Diamond et al., 2011*; *Navarro-Cano et al., 2015*) and the benefits for survival such as avoiding predators (*Slusarczyk, 1995*; *Kroon et al., 2008*; *Ji, 2011*) or intra- (*Tougeron et al., 2018*) and interspecific competition (*Dyugmedzhiev et al., 2019*), influence dormancy phenology at the species level. Thus, the life history hypothesis finds important support to explain the phenology of dormancy at different scales (e.g. the individual scale, differences between sexes). Although the energy limitation hypothesis seems to explain part of dormancy duration (*Pianka, 1970*; *Turbill and Prior, 2016*; *Wilsterman et al., 2021*), it cannot, by itself, explain the sex differences in dormancy phenology. In the next section, we examine how physiological constraints may be integrated into the life history hypothesis framework, that is, the trade-off between reproduction and survival, to explain the phenology of dormancy in endotherms and ectotherms.

## Dormancy and physiological constraints

From an energetic point of view, it is favorable to remain active as long as the energy balance is stable or positive, or conversely to remain in hibernation as long as the environment does not allow for a positive energy balance. Any deviation from this principle may highlight a balance rather in favor of reproduction (by sexual or non-sexual forms of natural selection) or survival (*Figure 1*) as expected by the life history hypothesis. Presumably, reducing the risk of extrinsic mortality favors dormancy while the environment allows a positive energy balance. In contrast, preparation for reproduction promotes activity while the environment does not allow a positive energy balance (*Snyder et al., 1961*). Thus, a dormancy phenology staggered with respect to the growing season (earlier emergence and immergence than expected) illustrates the selection pressure exerted by the trade-off between reproduction (earlier emergence than expected) and adult survival (earlier immergence than expected).

In our study, several elements might suggest that hibernation occurs even when environmental conditions allow for a positive energy balance. Gains in body mass observed for some individuals, even in species not known to hoard food, may indicate that the environment allows a positive energy balance for other individuals with comparable energy demands. In several species, females stay in hibernation (up to almost 2 months more) while males gain body mass of up to 9% after emergence,

**Table 2.** Species with dimorphisms biased in favor of males or females and their body mass gain during the year.
Body size dimorphism is calculated as male body size divided by female body size. See section "Sex differences in reproductive investment" for the determination of the body mass gain.

| Species | Body size dimorphism | Male body mass gain before mating (% of emergence body mass) | The end of reserve accumulation before hibernation for females |
|---|---|---|---|
| *Cricetus cricetus* | 1.14 | 9.35 | 27 days after male |
| *Cynomys leucurus* | 1.04 | 4.89 | 11 days after male |
| *Glis glis* | 0.97 | 6.63 | 14 days after male |
| *Microcebus murinus* | 0.96 | 9.01 | Same time as male |
| *Urocitellus parryii* | 0.97 | 0.49 | 35 days before male |

or one sex immerges while the second continues to accumulate energy reserves (*Table 2*). Sexual dimorphism may not be responsible for this sex differences as these observations concern species with a sexual size dimorphism biased towards both males or females (*Bintz, 1984Table 2*). To our knowledge, the only study that measured energy balance at the time of immergence showed that *Tamias striatus* (the eastern chipmunk) immerged while a positive energy balance could be maintained (*Humphries et al., 2002*). Observations of other species suggest immergence when little food is available, but supposedly enough to support activity (*Dobson et al., 1992*; *Grigg, 2000*; *Munro et al., 2008*; *Hoelzl et al., 2015*). On the contrary, years of low productivity can lead to later immergence (*Alcorn, 1940*; *O'Farrell et al., 1975*), probably due to a delay in the accumulation of reserves. This contradicts the view that hibernation duration should necessarily increase with energetic constraints.

In ectotherms as well, some observations confirmed a dormancy phenology staggered with respect to the growing season. In some reptiles and insects, individuals enter dormancy while ambient temperature and food were still high enough to promote activity (*Jameson, 1974*; *Jameson and Allison, 1976*; *Etheridge et al., 1983*; *Wiklund et al., 1996*). In *Elaphe obsoleta* (the black rat snake), part of the variation in emergence date was explained by the fact that smaller and younger individuals emerged later than others (*Blouin-Demers and Weatherhead, 2001*). This result is the opposite of what is expected from a thermoregulation perspective, since small individuals should reach their preferred temperature for activity more quickly (due to their low inertia) and should be the first to emerge (*Stevenson, 1985*). The authors proposed, on the contrary, that small individuals, subjected to a higher predation rate in spring, benefited from increased survival while remaining inactive. Males of *Vipera berus* (the common European adder) emerge before females in thermally unfavorable periods, leading to significant body mass loss because of the reproductive benefits of early emergence (*Herczeg et al., 2007*).

In the same way, the majority of insects enter dormancy long before environmental conditions deteriorate, and remain dormant sometimes long after favorable conditions return (*Tauber and Tauber, 1976*; *Kostál, 2006*). This strategy has been called 'temporal conservative bet-hedging' (*Hopper, 1999*). Temporal bet-hedging strategies reduce fitness variation across the years in a temporally fluctuating environment and supposedly result in higher average long-term fitness. In this case, all individuals in a population (conservative because of low phenotypic variability) reproduce only during the period that is always favorable through the years and avoid the period with adverse conditions in some years at the expense of possible reproductive benefits in years with favorable conditions. Temporal diversified bet-hedging exists in species for which the duration of dormancy varies within a single cohort (diversified because of high phenotypic variability) from one to several years (i.e. prolonged diapause), regardless of external conditions. Thus, whatever the environmental conditions, a small proportion of the progeny will experience optimal conditions to reproduce (*Hopper, 1999*).

The independence of ectotherm dormancy towards harsh environmental conditions is in contradiction with the vision of a passive inactivity induced by suboptimal temperature. Several physiological and behavioral thermoregulation mechanisms may facilitate entering into dormancy when the ambient temperature above ground is still high. Indeed, some ectotherms enter dormancy in summer (i.e. estivation or summer dormancy) and use deep burrows or crevices where the ambient temperature is much colder. Thus, by exploiting their habitat, some ectotherms are able to reduce their energy consumption (*Pinder and Storey, 1992*). On the other hand, some species are capable of an active

reduction in metabolism below that required under the simple passive effect of ambient temperature on metabolism (Q10 effect) (*Staples, 2016*). Ectotherm dormancy could therefore be less temperature dependent than previously thought and would allow survival under a wider spectrum of biotic and abiotic pressures.

## Study limits

Our statistical analysis were based on a limited number of hibernators, and we have accepted approximations concerning the phenology data for some species (e.g. date at which first/last individuals of each sex were observed) due to the limited data in the literature. The analysis included mainly Holarctic rodents except for *Erinaceus europaeus*, *Microcebus murinus*, and *Tachyglossus aculeatus*. Our analysis included two non-Holarctic species (*Microcebus murinus* and *Tachyglossus aculeatus*). Although they represent a very small minority of hibernating species, the results obtained seem to be consistent with Holarctic species. Hibernation in non-Holarctic species is supposed to have evolved in response to other environmental factors than food shortage, such as water shortage (*Bintz, 1984*; *R Development Core Team, 2019*; *Nowack et al., 2020*). However, similar selection pressures may therefore exist and should encourage further comparative research on hibernation in non-Holarctic and Holarctic species.

Bats were not included in this meta-analysis but represent an interesting model for hibernation biology, because the sex difference in reproduction phenology is very different from most hibernators. Thus, bats introduce varied and unique patterns to an understanding of hibernation phenology (*Willis, 2017*). In temperate bats, mating takes place just before hibernation during 'fall swarming' (*Thomas et al., 1979*). Females store sperm during winter and ovulation takes place shortly after emergence (*Buchanan, 1987*). In *Myotis lucifugus* (the little brown bat), males immerge after females, likely increasing their mating opportunities and subsequently recovering from body mass lost during mating (*Norquay, 2014*). Female bats likely emerged first because early parturition increased juvenile survival. The patterns observed were consistent with the life-history hypothesis. Although few data are currently available, future comparative studies between bat species should enhance our understanding of the life-history hypothesis. *Tamias sibericus* was also excluded from the analysis of immergence. Chipmunks are food-storing hibernators and wait until autumn to fill the hibernaculum with acorns (from masting in autumn) for winter feeding. Thus, reproductive effort may no longer have an impact on immergence at this time, but rather males may wait until all females hibernate in order to confirm the location of female burrows and increase mating success in the spring (*Kawamichi, 1996*). Thus, chipmunks may delay their immergence for reproductive benefits, in line with the life-history hypothesis. Although the analysis performed in this study only included some of hibernators, the available information suggested that the life history hypothesis may apply to the phenology of other hibernators.

## Conclusion

The sex difference in dormancy phenology observed in endotherms and ectotherms may be a widespread consequence of the trade-off between the benefits of being active for reproduction and the benefits of dormancy for survival (viz., the life-history hypothesis). Other non-exclusive and context-specific hypotheses have also been proposed (*Morbey and Ydenberg, 2001*) and further studies are needed to test them. Energy constraints explain a part of dormancy phenology in both endotherms and ectotherms (*Wilsterman et al., 2021*), but evidence from our study shows some independence of energy balance at the specific times of emergence and immergence into hibernation. This study shows that physiological constraints must be integrated into the survival-reproduction trade-off of hibernation phenology to be meaningful. Thus, we expect that dormancy phenology will exhibit multiple evolutionary causes, especially when many species are studied and compared. The occurance of dormancy at high altitudes and latitudes where few or no energy resources are available over part of the year appears to be supported for the physiological constraints hypothesis (*Ruf, 2012*), although this hypothesis appears to be of limited or partial importance in explaining the phenology of the transition from dormancy to activity and *vice-versa*. Dormancy in energetically benign periods, but unfavorable for reproduction, may be more widespread than previously thought. Such research highlights the opportunities of studying dormancy across a broad spectrum of species (*Wilsterman et al., 2021*).

## Materials and methods

### Review criteria

Our literature review was based on 152 hibernating species of mammals (see supplementary material 1 in *Constant et al., 2020*). In this study, we addressed what can be called ecological hibernation, i.e., the seasonal duration that an animal remains sequestered in its burrow or den, which is assumed to be directly linked to the reduced risk of predation. In contrast, we did not consider heterothermic hibernation, which corresponds to the time between the beginning and end of the use of torpor. So when we mention hibernation, emergence, or immergence, the specific reference is to ecological hibernation. We excluded non-seasonal hibernating species that do not have a consistent seasonal hibernation phenology (elephant shrew and marsupial species except *Burramys parvus* (the mountain pygmy possum)). We did not include species from the order *Carnivora* and *Chiroptera* because of a difference in reproductive phenology compared to the majority of other hibernators, especially due to delayed embryo implantation (*Sandell, 1990*). This implies different trade-offs between hibernation and reproduction that require separate analysis. In addition, there were few data available on both reproduction and hibernation for hibernating bat species (see below for traits needed for inclusion; also the Discussion for hypotheses applied to these groups). Only one bird species is considered to be a hibernator, and no information is available on sex differences in hibernation phenology (*Woods, 2004*; *Woods et al., 2019*).

Each of the following literature reviews was conducted using the search engine Google Scholar with specific keywords and considered articles up to and including January 2021. In total, our literature search allowed the inclusion of 29 hibernating mammals in the analysis for which we have both reproduction and hibernation data including mainly rodents, a monotreme, a primate, and an *Eulipotyphla* species.

### Sex difference in hibernation phenology

We searched for hibernation phenology for each sex based on the average date of emergence and immergence in the same population. When these types of data were not available, we accepted the date at which the first/last individuals of each sex were observed or the approximate sex difference available in the text. The search criteria were based on combining the following terms: (scientific OR common names of species) AND (phenology OR annual cycle OR hibernation). Because of their imprecision, we excluded the studies for which hibernation season phenology was deduced from the presence of active individuals on a monthly basis. Because of their imprecision, we excluded the studies for which hibernation season phenology was deduced from the presence of active individuals on a monthly basis. This excluded four studies (*Dunford, 1974*; *Gashwiler, 1976*; *Mouhoub Sayah et al., 2009*; *Randrianambinina et al., 2003*). As the data were averaged for each species (see section 'Statistics') we did not use data with exceptional variation between years within the same study site. This excluded data from *Munroe, 2011* on the sex difference in immergence date (55 days difference between the two years) for *Xerospermophilus tereticaudus* (the round-tailed ground squirrel). *Otospermophilus beecheyii* (the California ground squirrel) appeared to be a species with great variation in hibernation phenology and whether males and females hibernated (*Dobson and Davis, 1986*; *Holekamp et al., 1988*). These data were therefore not included in this study.

From the remaining data, we calculated protandry and the sex difference in immergence into hibernation as female Julian date – male Julian date.

We have excluded immergence data for *Tamias sibericus* because this species does not follow the general phenology of other species (see 'Study limitations' section for details).

### Sex differences in reproductive investment and sexual dimorphism

For males, relative body mass changes during the mating period (hereafter referred to as 'Δ body mass during mating') was calculated as follows: (body mass at the end of mating - body mass at the beginning of mating)/body mass at the beginning of mating. Male immergence is expected to be constrained by the long-term negative effects of reproductive stress (*Millesi et al., 1998*). In addition to the energetic costs of mating, some recently emerged males lost body mass before females emerged from hibernation, which may have resulted in physiological stress. Thus, we calculated relative body mass change between emergence and the end of mating, hereafter referred to as 'Δ body

mass through the end of mating' as follows: (body mass at the end of mating - body mass at emergence)/body mass at emergence.

For all data on changes in body mass, the search was conducted by combining the following terms: (scientific OR common names of species) AND (body mass change OR annual body mass). To be as accurate as possible, we have obtained data only when measured at the same or nearby the study site that was used for hibernation data. In cases where information were not directly available in the text or table, we used the software Plot Digitizer (*Huwaldt and Steinhorst, 2015*) to extract the data from graphs. This software has recently been validated for this use (*Aydin and Yassikaya, 2021*). The start and end dates of mating were estimated from information available in the text or from other studies at the same study site. When the mating period could not be clearly determined, studies were omitted from analysis.

Maternal effort may have constrained female immergence in two ways: the duration and energy cost of maternal care. Maternal effort duration was calculated as the sum of the gestation and lactation periods. Energy cost of reproductive effort also known as 'specific reproductive effort' is calculated as the mean annual litter size times juvenile body mass at weaning divided by the body mass of adult females before reproduction (*Armitage, 1981*). For the analysis, we give priority to reproductive and body mass data from the same or very close study sites as the hibernation phenology data. We made a few exceptions for juvenile body mass at weaning, for which some data were obtained in *Hayssen, 2008*, male reproductive effort for *Spermophilus brunneus* and *Zapus princeps*, and the length of gestation and lactation. Maternal effort duration is calculated as the sum of the gestation and lactation periods. We obtained data on the length of gestation and lactation from the AnAge database (The Animal Aging and Longevity Database; *de Magalhães and Costa, 2009*), and complemented these data with a specific search combining the following terms: (scientific OR common names of species) AND (lactation duration OR gestation duration). As maternal effort duration and lactation data were not available in the literature for *Urocitellus brunneus* and *Ictidomys parvidens*, respectively, we used averages for the clade *Marmotini* obtained from *Hayssen, 2008*. The body mass at weaning for *Urocitellus brunneus* corresponds to the body mass at weaning of *Urocitellus townsendii* (phylogenetically close species) corrected for the difference in body mass of the females from *Hayssen, 2008* as body mass accounts for 76–84% of the variation in weaning mass in the clade *Marmotini* (*Hayssen, 2008*).

## Climate data

Species living in harsh conditions may be constrained by a shorter active season that might influence sex differences in hibernation phenology. To take this into account in the models (see section 'Statistics'), the location (latitude and longitude) of hibernation study sites were recorded, and when not provided we determined their location using Google Earth and the location description. Then the location data were used to extract values of the mean temperature of the coldest month (known as 'BIO6' in the database and hereafter referred to as minimum temperature) and annual precipitation (known as 'BIO12' in the database) from an interpolated climate surface (BIOCLIM) with 1 km² resolution (30 s) based on data for the period 1970–2000 (*Hijmans et al., 2005*).

## Statistics

We used phylogenetic generalized least squares (PGLS) models (see *Constant et al., 2020* for details and *Figure 6* for consensus phylogenetic trees) to account for the non-independence of phylogeny-related species with the 'ape 5.0,' 'apTreeshape 1.5,' and 'caper 1.0' packages in R v. 3.6.2 (*Orme et al., 2013*; *Paradis, 2011*; *Paradis and Schliep, 2019*; *R Development Core Team, 2019*). Each PGLS model produces a $\lambda$ parameter representing the effect of phylogeny ranging between 0 (no phylogeny effect) and 1 (covariance entirely explained by co-ancestry).

The PGLS models used only one value per species for each factor. For hibernation phenology, body mass, reproductive effort, and climatic variable, we first averaged by study when data were available over several years, and then we averaged the data for the species. This produces equal weighting between studies on the same species.

To test the life history and physiological constraint hypotheses on emergence, protandry was the dependent variable while, Δ body mass during mating, male body mass at emergence, sexual dimorphism at emergence, minimum temperature, annual precipitation, food-storing, and late mating were independent variables. The 'late mating' factor aims to test for lower protandry when

mating was more greatly delayed after the onset of the annual activity period, as has been shown for reptiles (*Olsson et al., 1999*), hereafter referred to « delay in mating ». The 'food-storing' factor tested for lower protandry for species that store food in a burrow and consume it after the last torpor bouts, which may allow them to prepare for reproduction without emerging above ground (*Williams et al., 2014*). Food-storing species have been identified in several studies (*Kenagy et al., 1989*; *Wall and Stephen, 1990*; *Michener, 1992*; *Bieber, 2004*). To test the life history and physiological constraint hypotheses on immergence, sex difference in immergence was the dependent variable while Δ body mass through the end of mating, duration of maternal effort, specific reproductive effort of female, sexual dimorphism in immergence, minimum temperature and annual precipitation were independent variables. In both models, we tested for a 'two-factor interaction' between either minimum temperature, annual precipitation male, and female reproductive effort. were independent variables.

The two models are described hereafter (see *Figure 3—source data 1*, *Figure 4—source data 1* and *Figure 5—source data 1* for datasets):

Protandry ~late_mating + food-storing + male body mass +dimorphism body mass at emergence + Δ body mass during mating * Minimum temperature + Δ body mass during mating * annual precipitation

Sex difference in immergence ~log(specific reproductive effort) * Minimum temperature + log(specific reproductive effort) * precipitation + Δ body mass through the end of mating * Minimum temperature + Δ body mass through the end of mating * Precipitation + Maternal effort duration* Precipitation +Maternal effort duration * Minimum temperature + Body mass immergence + Dimorphisme at immergence

For both models, we used the dredge function of the MuMIn package (version 1.43.17; *Barto, 2020*) to select the best model based on the corrected Akaike information criterion (AICc). Normality and homoscedasticity were checked by graphical observation. We tested for multicollinearity using variance inflation factors (we required VIF <3) on linear models including the factors of the best models. PGLS models do not include calculations of VIFs (*Wartel et al., 2019*; *Ancona et al., 2020*). Female-specific reproductive effort was log-transformed to improve the fit to the normality of the residuals. All independent variables were standardized (using *z-scores*) in multi-factor models, so that their coefficients are directly comparable as estimates of effect sizes (*Abdi, 2007*).

## Acknowledgements

The authors are grateful to the researchers for providing specific information on previous studies including published data, in particular to Carina Siutz, Danielle Levesque, Andrey Tchabovsky, Eric Rickart, Philip Leitner, and John Hoogland for sharing data, some of which have been used in the meta-analysis of this article. The authors are also grateful to John Drake and two referees who provided excellent suggestions for manuscript improvements. SG was financially supported by the Austrian Science Fund (FWF, Grant No. P31577-B25) and the Austrian Agency for International Cooperation in Education and Research (OeAD – Scientific and Technological Cooperation, Grant No. FR 09/2020).

## Additional information

### Funding

| Funder | Grant reference number | Author |
| --- | --- | --- |
| Austrian Science Fund | FWF Grant No. P31577-B25 | Sylvain Giroud |
| Austrian Agency for International Cooperation in Education and Research | OeAD - Scientific and Technological Cooperation Grant No. FR 09/2020 | Sylvain Giroud |

The funders had no role in study design, data collection and interpretation, or the decision to submit the work for publication.

## Author contributions

Théo Constant, Conceptualization, Data curation, Formal analysis, Methodology, Writing – original draft, Writing – review and editing, Project administration; F Stephen Dobson, Data curation, Formal analysis, Supervision, Validation, Methodology, Writing – original draft, Writing – review and editing; Caroline Habold, Sylvain Giroud, Resources, Supervision, Funding acquisition, Validation, Writing – original draft, Project administration, Writing – review and editing

## Author ORCIDs

Théo Constant ⓘ https://orcid.org/0000-0002-7166-8273
Caroline Habold ⓘ http://orcid.org/0000-0002-6881-6546
Sylvain Giroud ⓘ http://orcid.org/0000-0001-6621-7462

Reviewer #2 (Public review): https://doi.org/10.7554/eLife.89644.3.sa1
Author response https://doi.org/10.7554/eLife.89644.3.sa2

---

# Additional files

## Supplementary files

• MDAR checklist

## Data availability

The data and computer code supporting the results are available at https://github.com/Theo-Constant/Evolutionary-trade-offs-in-dormancy-phenology (copy archived at *Constant and Pansanel, 2024*).

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
