## [Editor Report · eLife assessment]

This **valuable** and ambitious review examines seasonal dormancy in various species, including hibernating mammals (excluding bats and bears) and ectotherms. It provides a **solid** test of hypotheses on dormancy timing, considering energetic constraints and life history as alternative drivers. The review will be of interest to evolutionary biologists.

---

## [Referee Report · Reviewer #2 (Public review)]

Summary:

An article with lots of interesting ideas and questions regarding the evolution of timing of dormancy, emphasizing mammalian hibernation but also including ectotherms. The authors compare selective forces of constraints due to energy availability versus predator avoidance and requirements and consequences of reproduction in a review of between and within species (sex) differences in the seasonal timing of entry and exit from dormancy.

Strengths:

The multispecies approach including endotherms and ectotherms is ambitious. This review is rich with ideas if not in convincing conclusions. Limitations are discussed yet are impactful, namely that differences among and within species are contrast only for ecological hibernation (the duration of remaining sequestered) and not for "heterothermic hibernation" the period between first and last torpor. Differences between the two can have significant energetic consequences, especially for mammals returning to euthermic levels of body temperature whilst remaining in their cold burrows before emerging, eg. reproductively developing males in spring.

Weaknesses:

The differences between physiological requirements for gameatogenesis between sexes that affect the timing of heterothermy and need for euthermy during mammalian hibernator are significant issues that underlie, but are under discussed, in this contrast of selective pressures that determine seasonal timing of dormancy. Some additional discussion of the effects of rapid rapid climate change on between and within species phenologies of dormancy would have been interesting.

---

## [Author Response]

The following is the authors’ response to the original reviews.

**Public Reviews**

**Reviewer #1 (Public Review):**
Summary:Dormancy/diapause/hibernation (depending on how the terms are defined) is a key life history strategy that allows the temporal escape from unfavorable conditions. Although environmental conditions do play a major role in inducing and terminating dormancy (authors call this energy limitation hypothesis), the authors test a mutually non-exclusive hypothesis (life-history hypothesis) that sex-specific selection pressures, at least to some extent, would further shape the timing of these life-history events. Authors use a metanalytic approach to collect data (mainly on rodents) on various life-history traits to test trade-offs among these traits between sexes and how they affect entry and termination of dormancy.Strengths:I found the theoretical background in the Introduction quite interesting, to the point and the arguments were well-placed. How sex-specific selection pressures would drive entry and termination of diapause in insects (e.g. protandry), especially in temperate butterflies, is very well investigated. Authors attempt to extend these ideas to endotherms and trying to find general patterns across ectotherms and endotherms is particularly exciting. This work and similar evidence could make a great contribution to the life-history theory, specifically understanding factors that drive the regulation of life cycle timing.Weaknesses:(1) I felt that including 'ectotherms' in the title is a bit misleading as there is hardly (in fact any?) any data presented on ectotherms. Also, most of the focus of the discussion is heavily mammal (rodent) focussed. I believe saying endotherms in the title as well is a bit misleading as the data is mammalfocused.

We change the title to : "Evolutionnary trade-offs in dormancy phenology". This is a hybrid article comprising both a meta-analysis and a literature review. Each of these parts brings new elements to the hypotheses presented. The statistical analyses only concern mammals and especially rodent species. But the literature review highlighted links between the evolution of dormancy in ectotherms and endotherms that have not been linked in previous studies. We feel it is important for readers to know that much of the discussion will focus on the comparison of these two groups. But we understand that placing the term ectotherms in the title might suggest a meta-analysis including these two groups.

In addition, we indicated more specifically in the abstract and at the end of the introduction that the article includes two approaches associated with different groups of animals.

We also specified in the section « review criteria » that:

Only one bird species is considered to be a hibernator, and no information is available on sex differences in hibernation phenology (Woods and Brigham 2004, Woods et al. 2019).

We have also added a "study limitations" section, which explains that although the meta-analysis is limited by the data available in the literature, the information available for the species groups not studied seems to support our results.

(2) I think more information needs to be provided early on to make readers aware of the diversity of animals included in the study and their geographic distribution. Are they mostly temperate or tropical? What is the span of the latitude as day length can have a major influence on dormancy timings? I think it is important to point out that data is more rodent-centric. Along the line of this point, is there a reason why the extensively studied species like the Red Deer or Soay Sheep and other well-studied temperate mammals did not make it into the list?

We specified in the abstract and at the end of the introduction that the species studied in the metaanalysis are mainly Holarctic species. We have also added a map showing all the study sites used in the meta-analysis. Finally, we've noted in the methods and added a "study limitation" section at the end of the discussion an explanation for those species that were not studied in the meta-analysis and the consequences for the interpretation of results

The hypotheses developed in this article are based on the survival benefits of seasonal dormancy thanks to a period of complete inactivity lasting several months. The Red Deer or Soay Sheep remain active above ground throughout the year.

The effect of photoperiod on phenology is one of the mechanisms that has evolved to match an activity with the favorable condition. In this study, we are not interested in the mechanisms but in the evolutionary pressures that explain the observed phenology. Interspecific variation in the effect of photoperiod results from different evolutionary pressures, which we are trying to highlight. It is therefore not necessary to review mechanisms and effects of photoperiod, themselves requiring a lengthy review.

We also tested the “physiological constraint hypothesis” on several variables. Temperature and precipitation are factors correlated with sex differences in phenology of hibernation. These factors allow consideration of the geographical differences that influence hibernation phenology.

(3) Isn't the term 'energy limitation hypothesis' which is used throughout the manuscript a bit endotherm-centric? Especially if the goal is to draw generalities across ectotherms and endotherms. Moreover, climate (e.g. interaction of photoperiod and temperature in temperatures) most often induces or terminates diapause/dormancy in ectotherms so I am not sure if saying 'energy limitation hypothesis' is general enough.

We renamed this hypothesis the "physiological constraint hypothesis" and we have made appropriate changes in the text so as not to focus physiological constraints solely on energy aspects.

(4) Since for some species, the data is averaged across studies to get species-level trait estimates, is there a scope to examine within population differences (e.g. across latitudes)? This may further strengthen the evidence and rule out the possibility of the environment, especially the length of the breeding season, affecting the timing of emergence and immergence.

For a given species, data on hibernation phenology are averaged for different populations, but also for the same population when measurements are taken over several years. To test these hypotheses on a population scale, precise data on reproductive effort would be needed for each population tested, but this concerns very few species (less than 5).

Testing the effects of temperature and precipitation allows us to take into account the effects of climate on phenology.

(5) Although the authors are looking at the broader patterns, I felt like the overall ecology of the species (habitat, tropical or temperate, number of broods, etc.) is overlooked and could act as confounding factors.

Yes, that's why we also tested the physiological constraints hypothesis, including the effect of temperature and precipitation. For the life-history hypothesis, we also tested reproductive effort, which takes into account the number of offspring per year.

(6) I strongly think the data analysis part needs more clarity. As of now, it is difficult for me to visualize all the fitted models (despite Table 1), and the large number of life-history traits adds to this complexity. I would recommend explicitly writing down all the models in the text. Also, the Table doesn't make it clear whether interaction was allowed between the predictors or not. More information on how PGLS were fitted needs to be provided in the main text which is in the supplementary right now. I kept wondering if the authors have fit multiple models, for example, with different correlation structures or by choosing different values of lambda parameter. And, in addition to PGLS, authors are also fitting linear regressions. Can you explain clearly in the text why was this done?

To simplify the results, we reduced the number of models to just three: one for emergence and two for immergence. In place of Table 1, we have written the structure of the models used.We have added a sentence to the statistics section: “each PGLS model produces a λ parameter representing the effect of phylogeny ranging between 0 (no phylogeny effect) and 1 (covariance entirely explained by co-ancestry)”. We have tested only three PGLS models and the estimated lambda value for these models is 0.

(7) Figure 2 is unclear, and I do not understand how these three regression lines were computed. Please provide more details.

We tested new models and modified existing figures.

**Reviewer #2 (Public Review):**
Summary:An article with lots of interesting ideas and questions regarding the evolution of timing of dormancy, emphasizing mammalian hibernation but also including ectotherms. The authors compare selective forces of constraints due to energy availability versus predator avoidance and requirements and consequences of reproduction in a review of between and within species (sex) differences in the seasonal timing of entry and exit from dormancy.Strengths:The multispecies approach including endotherms and ectotherms is ambitious. This review is rich with ideas if not in convincing conclusions.Weaknesses:The differences between physiological requirements for gameatogenesis between sexes that affect the timing of heterothermy and the need for euthermy during mammalian hibernator are significant issues that underlie but are under-discussed, in this contrast of selective pressures that determine seasonal timing of dormancy. Some additional discussion of the effects of rapid climate change on between and within species phenologies of dormancy would have been interesting.
**Reviewer #2 (Recommendations For The Authors):**
This review provides a very interesting and ambitious among and within-species comparison of the seasonal timing of entry and exit from dormancy, emphasizing literature from hibernating mammals (sans bats and bears) and with attention to ectotherms. The authors test hypotheses related to the timing of food availability (energy) versus life history considerations (requirements for reproduction, avoiding predation) while acknowledging that these are not mutually exclusive. I offer advice for clarifications and description of the limitations of the data (accuracy of emergence and immergence times), but mainly seek more emphasis for small mammalian hibernators on the contrast for requirements for significant periods of euthermy prior to the emergence in males versus females, a contrast that has energetic and timing consequences in both the active and hibernation seasons.A consideration alluded to but not fully explained or discussed is the differences in mammals between species and sexes in the timing of what can be called ecological hibernation, which is the seasonal duration that an animal remains sequestered in its burrow or den, and heterothermic hibernation, between the beginning and end of the use of torpor. The two are not synonymous. When "emergence" is the first appearance above ground, there is a significant missing observation key to the energetic contrasts discussed in this review, that of this costly pre-emergence behavior.

To explain the difference between heterothermic hibernation and ecological hibernation, we've added a section in review Criteria from materials and methods :

“In this study, we addressed what can be called ecological hibernation, i.e. the seasonal duration that an animal remains sequestered in its burrow or den, which is assumed to be directly linked to the reduced risk of predation. In contrast, we did not consider heterothermic hibernation, which corresponds to the time between the beginning and end of the use of torpor. So when we mention hibernation, emergence or immergence, the specific reference is to ecological hibernation.”

In arctic and other ground squirrel species, males remain at high body temperatures after immerging and remaining in their burrows in the fall for several days to a week, and more consistently and importantly, males that will attempt to breed in the spring end torpor but remain constantly in their burrows for as much as one month at great expense whilst undergoing testicular growth, spermatogenesis, spemiation, and sperm capacitation, processes that require continuous euthermy. Female arctic ground squirrels and non-breeding males do not and typically enter their first torpor bout 1-2 days after immergence and first appear above ground 1-3 days after their last arousal in spring.The weeks spent euthermic in a cold burrow in spring by males while undergoing reproductive maturation require a significant energetic investment (can equate to the cost of the previous heterothermic period) that contrasts profoundly with the pre-mating energetic investment by females.Males cache food in their hibernacula and extend their active season in late summer/fall in order to do so and feed from these caches in spring after resuming euthermy, often emerging at body weights similar to that at immergence. Similar between-sex differences in the timing of hibernation and heterothermy occur in golden-mantled and Columbian ground squirrels and likely most other Urocitellus spp., though less well described in other species. These differences are related to life histories and requirements for male vs. female gameatogenesis and, at the same time, energetic considerations in the costs to males for remaining euthermic while undergoing spermatogenesis and the cost related to whether males undergo gonadal development being dependent on individual body mass and cache size. These issues should be better discussed in this review.It is the time required to complete spermatogenesis, spermiation, and maturation of sperm not the time for growth of different sizes of testes that drives the preparation time for males. This is relatively constant among rodents. I challenge the assumption that larger testes take longer to grow than smaller ones.

We took this comment into account. As we found little evidence of an increase in testicular maturation time with relative testicular size (apart from table 4 in Kenagy and Trombulak, 1986), we no longer tested the effect of relative testicular size on protandry.

We examined whether the ability to store food before hibernation might reduce protandry. Although food storage in the burrow may be favored for overcoming harsh environments or predation, model selection did not retain the food-storing factor. Thus, the ability to accumulate food in the burrow was not by itself likely to keep males of some species from emerging earlier (e.g. Cricetus cricetus, protandry : 20 day, Siutz et al., 2016). Early emerging males may benefit from consuming higher quality food or in competition with other males (e.g., dominance assertion or territory establishment, Manno and Dobson 2008).

We developed these aspects in the discussion

While it is admirable to include ectotherms in such a broad review and modelling, I can't tell what data from how many ectothermic species contributed to the models and summary data included in the figures.

Too few data on ectotherms were available to include ectotherms in the meta-analysis

Some consideration should be made to the limitations of the data extracted from the literature of the accuracy of emergence and immergence dates when derived from only observations or trapping data. The most accurate results come from the use of telemetry for location and data logging reporting below vs. above ground positioning and body temperature.

We added a "study limits" section to the discussion to address all the limits in this commentary.

L64 "favor reproduction", better to say "allow reproduction", since there is strong evolutionary pressure to initiate reproduction early, often anticipating favorable conditions for reproduction, to maximize the time available for young to grow and prepare for overwintering themselves.Also, generally, it is not how "harsh" an environment is but rather how short the growing season is.

We took this comment into account.

L80 More simply, individuals that have amassed sufficient energy reserves as fat and caches to survive through winter may opt to initiate dormancy. This may decrease but not obviate predation, since hibernating animals are dug from their burrows and eaten by predators such as bears and ermine.

In this sentence, we indicated a gap between dormancy phenology and the growing season, which suggests survival benefits of dormancy other than from a physiological point of view. We've changed the sentence to make it clearer : “However, some animals immerge in dormancy while environnemental conditions would allow them (from a physiological point of view) to continue their activity, suggesting other survival benefits than coping with a short growing season”

L88 other physiological or ecological factors.... (gameatogenesis).

In this study, we examine possible evolutionary pressures and therefore the environmental factors that may influence hibernation phenology. We focus on reproductive effort because, assuming predation pressure, we would expect a trade-off between survival and reproduction.

L113 beginning early to afford long active seasons to offspring while not compromising the survival of parents.

We added to the sentence:

“For females, emergence phenology may promote breeding and/or care of offspring during the most favorable annual period (e.g., a match of the peak in lactational energy demand and maximum food availability, Fig. 1) or beginning early to afford long active seasons to offspring while not compromising the survival of parents.”

L117 based on adequate preparation for overwintering and enter dormancy....

We modified the sentence as follows :

recovering from reproduction, and after acquiring adequate energy stores for overwintering”

L123 given that males outwardly invest the least time in reproduction yet generally have shorter hibernation seasons would seem to reject this hypothesis. This changes if you overtly include the time and energy that males expend while remaining euthermic preparing for hibernation, a cost that can be similar to energy expended during heterothermy.

Males invest a lot of time in reproduction before females emerge (whether for competition or physiological maturation) and some males seem to be subject to long-term negative effects linked to reproductive stress (see Millesi, E., Huber, S., Dittami, J., Hoffmann, I., & Daan, S. (1998). Parameters of mating effort and success in male European ground squirrels, Spermophilus citellus. Ethology, 104(4), 298-313). Both processes may contribute to reducing the duration of male hibernation.

L125 again, costs to support euthermy in males undergoing reproductive development is an investment in reproduction.

You're right, but it's difficult to quantify. We tested a model that takes into account the reproductive effort during reproduction and prior to reproduction. We also considered the hypothesis that species living in a cold climate might have a low protandry while having a high reproductive effort due to their ability to feed in the burrow (interaction effect between reproductive effort and temperature). We think these changes answer your comment.

L134 It isn't growing large testes that takes time, but instead completing spermatogenesis and maturation of sperm in the epdidymides.

We removed this part.

L140 Later immergence in male ground squirrels is related to accumulation and defense of cached food, activities that are related to reproduction the next spring. An experimental analysis that would be revealing is to compare immergence times in females that completed lactation to the independence of their litters vs. females that did not breed or lost their litters. Who immerges first?Body mass variation from emergence to the end of mating in males seems to explain the delayed immergence of males in species that don't hide food in their burrows for hibernation. For example, in spermophilus citellus, males immege on average more than 3 weeks after females, yet they do not hide food in their burrows for the winter.

Such a study already exists and shows that non-breeding females immerge earlier than breeding females. We refer to it

L386: “In mammals, males and females that invest little or not at all in reproduction exhibit advances in energy reserve accumulation and earlier immergence for up to several weeks, while reproductive congeners continue activity (Neuhaus 2000, Millesi et al. 2008a).”

L164 So you examined literature from 152 species but included data from only 29 species? Did you include data from social hibernators (marmots) that mate before emergence?

With current models, we have 28 different species. We have few species because very few have data on both sex difference data and information on reproductive effort data (especially for males).

Data on sex differences in hibernation were not available for social hibernating species.

L169 Were these data from trapping or observation results? How reliable are these versus the use of information from implanted data loggers or collars that definitively document when euthermy is resumed and/or when immergence and first emergence occurs (through light loggers)?

We did not focus heterothermic hibernation, but in ecological hibernation. We have no idea of the margin of error for these types of data, but we have discussed these limitations in the "Study limitations" section.

L180, again, it is the time required to complete spermatogenesis and spermiation not the time for the growth of different sizes of testes that drives the preparation time for males. This is relatively constant among rodents. I challenge the assumption that larger testes take longer to grow than smaller ones.

We removed this part.

L200 Males that accumulate caches in fall and then feed from those during the spring pre-emergence euthermic interval and after will often be at their seasonal maximum in body mass. Declining from that peak may not be stressful.

It has been suggested that reproductive effort in Spermophilus citellus might induce long-term negative effects that delay male immergence.

Millesi, E., Huber, S., Dittami, J., Hoffmann, I., & Daan, S. (1998). Parameters of mating effort and success in male European ground squirrels, Spermophilus citellus. Ethology, 104(4), 298-313.

L210 How about altitude, which affects the length of the growing season at similar latitudes?

We extracted the location of each study site to determine the temperature and precipitation at that precise location (based on interpolated climate surface). We therefore take into account differences in growing season (based on temperature) in altitude between sites.

L267 How did whether males cache food or not figure into these comparisons? Refeeding before mating occurs during the pre-emergence euthermic interval.

We removed this part.

L332, 344 not a "proxy" but functionally related to advantages in mating systems with multiple mating males.

We removed this part.

L353 The need for a pre-emergence euthermic interval in male ground squirrels requires costs in the previous active season in accumulating and defending a cache and the proximal costs in spring while remaining at high body temperatures prior to emergence with resulting loss in body mass or devouring of the cache.

You're right, but in this section, we quickly explain the benefits of food catching compared with other species that don't do so.

L385 This review should discuss why females are not known to cache and contrast as "income breeders" from "capital breeder" males. What advantages of caches are females indifferent to (no need for a prolonged pre-emergence period) and what costs of accumulating caches do they avoid (prolonged activity period and defense of caches).

We clarified the case of female emergence.

L321 : “Thus, an early emergence of males may have evolved in response to sexual selection to accumulate energy reserve in anticipation of reproductive effort. Females, on the contrary, are not subject to intraspecific competition for reproduction and may have sufficient time before (generally one week after emergence) and during the breeding period to improve their body condition.”

L388 I don't understand the logic of the conclusion that "did not ...adequately explain the late male immergence" in this section. The greater mass loss in males over the mating period is afforded by the presence of a cache that requires later immergence.

We removed this part.

L412 Not just congeners that invest less in reproduction, but within species individuals that do not attempt to breed in one or more years and thus have no reproductive costs should be an interesting comparison for differences in phenology from individuals that do breed. Non-breeders are often yearlings but can be a significant overall proportion of males that fail to fatten or cache enough to afford a pre-emergence euthermic period.

L385: “In mammals, males and females that invest little or not at all in reproduction exhibit advances in energy reserve accumulation and earlier immergence for up to several weeks, while reproductive congeners continue activity (Neuhaus 2000, Millesi et al. 2008a).”

The sentence refers to individuals who reproduce little or not at all.

L445 Males that gain weight between emergence and mating may do so by feeding from a cache regardless of how "harsh" an environment is.

We observe this phenomenon even in species that are not known to hoard food

“Gains in body mass observed for some individuals, even in species not known to hoard food, may indicate that the environment allows a positive energy balance for other individuals with comparable energy demands.”

L492 Some insects retreat to refugia in mid-summer to avoid parasitism (Gynaephora).

Escape from parasites is also a benefit of dormancy.

Fig 1 - It is difficult to see the differences in black and green colors, esp if color blind.Maternal effort is front-loaded within the active season (line for "optimal period" shown in midseason).Add "energy" underneath (c) Prediction (H1) and "reproduction" underneath (d) "Prediction (H2). Explain the orange vs black, green colors of triangles.

We made the necessary changes

Fig 2 - I don't buy the regression lines as significant in this figure. The red line, cannot have a regression with two sample points and without the left-hand most dot, nothing is significant.

We deleted this graph.

Fig 3 - females only?

We deleted this graph.